# Inference Aided Reinforcement Learning for Incentive Mechanism Design in Crowdsourcing

**Zehong Hu**
Alibaba Group, Hangzhou, China
HUZE0004@e.ntu.edu.sg

**Yitao Liang**
University of California, Los Angeles
yliang@cs.ucla.edu

**Jie Zhang**
Nanyang Technological University
ZhangJ@ntu.edu.sg

**Zhao Li**
Alibaba Group, Hangzhou, China
lizhao.lz@alibaba-inc.com

**Yang Liu**
University of California, Santa Cruz/Harvard University
yangliu@ucsc.edu

## Abstract

Incentive mechanisms for crowdsourcing are designed to incentivize financially self-interested workers to generate and report high-quality labels. Existing mechanisms are often developed as one-shot static solutions, assuming a certain level of knowledge about worker models (expertise levels, costs of exerting efforts, etc.). In this paper, we propose a novel inference aided reinforcement mechanism that learns to incentivize high-quality data sequentially and requires no such prior assumptions. Specifically, we first design a Gibbs sampling augmented Bayesian inference algorithm to estimate workers' labeling strategies from the collected labels at each step. Then we propose a reinforcement incentive learning (RIL) method, building on top of the above estimates, to uncover how workers respond to different payments. RIL dynamically determines the payment without accessing any ground-truth labels. We theoretically prove that RIL is able to incentivize rational workers to provide high-quality labels. Empirical results show that our mechanism performs consistently well under both rational and non-fully rational (adaptive learning) worker models. Besides, the payments offered by RIL are more robust and have lower variances compared to the existing one-shot mechanisms.

## 1 Introduction

The ability to quickly collect large-scale and high-quality labeled datasets is crucial for Machine Learning (ML). Among all proposed solutions, one of the most promising options is crowdsourcing [7, 19, 4, 18]. Nonetheless, it has been noted that crowdsourced data often suffers from quality issue, due to its salient feature of no monitoring and no ground-truth verification of workers' contribution. This quality control challenge has been attempted by two relatively disconnected research communities. From the more ML side, quite a few inference techniques have been developed to infer true labels from crowdsourced and potentially noisy labels [16, 11, 24, 23]. These solutions often work as one-shot, post-processing procedures facing a static set of workers, whose labeling accuracy is fixed and *informative*. Despite their empirical success, the aforementioned methods ignore the effects of *incentives* when dealing with human inputs. It has been observed both in theory and practice that, without appropriate incentive, selfish and rational workers tend to contribute low quality, uninformative, if not malicious data [17, 12]. Existing inference algorithms are very vulnerable to

these cases - either much more redundant labels would be needed (low quality inputs), or the methods would simply fail to work (the case where inputs are uninformative and malicious).

From the less ML side, the above quality control question has been studied in the context of *incentive mechanism design*. In particular, a family of mechanisms, jointly referred as *peer prediction*, have been proposed [15, 8, 21, 2]. Existing peer prediction mechanisms focus on achieving *incentive compatibility* (IC), which is defined as that truthfully reporting private data, or reporting high quality data, maximizes workers' expected utilities. These mechanisms achieve IC via comparing the reports from the to-be-scored worker, against those from a randomly selected reference worker, to bypass the challenge of no ground-truth verification. However, we note several undesirable properties of these methods. Firstly, from learning's perspective, collected labels contain rich information about the ground-truth labels and workers' labeling accuracy. Existing peer prediction mechanisms often rely on reported data from a small subset of reference workers, which only represents a limited share of the overall collected information. In consequence, the mechanism designer dismisses the opportunity to leverage learning methods to generate a more credible and informative reference answer for the purpose of evaluation. Secondly, existing peer prediction mechanisms often require a certain level of prior knowledge about workers' models, such as the cost of exerting efforts, and their labeling accuracy when exerting different levels of efforts. However, this prior knowledge is difficult to obtain under real environment. Thirdly, they often assume workers are all fully rational and always follow the utility-maximizing strategy. Rather, they may adapt their strategies in a dynamic manner.

In this paper, we propose an *inference-aided reinforcement mechanism*, aiming to merge and extend the techniques from both inference and incentive design communities to address the caveats when they are employed alone, as discussed above. The high level idea is as follows: we collect data in a sequential fashion. At each step, we assign workers a certain number of tasks and estimate the true labels and workers' strategies from their labels. Relying on the above estimates, a reinforcement learning (RL) algorithm is proposed to uncover how workers respond to different levels of offered payments. The RL algorithm determines the payments for the workers based on the collected information up-to-date. By doing so, our mechanism not only incentivizes (non-)rational workers to provide high-quality labels but also dynamically adjusts the payments according to workers' responses to maximize the data requester's cumulative utility. Applying standard RL solutions here is challenging, due to unobservable states (workers' labeling strategies) and reward (the aggregated label accuracy) which is further due to the lack of ground-truth labels. Leveraging standard inference methods seems to be a plausible solution at the first sight (for the purposes of estimating both the states and reward), but we observe that existing methods tend to over-estimate the aggregated label accuracy, which would mislead the superstructure RL algorithm.

We address the above challenges and make the following contributions: (1) We propose a Gibbs sampling augmented Bayesian inference algorithm, which estimates workers' labeling strategies and the aggregated label accuracy, as done in most existing inference algorithms, but significantly lowers the estimation bias of labeling accuracy. This lays a strong foundation for constructing correct reward signals, which are extremely important if one wants to leverage reinforcement learning techniques. (2) A reinforcement incentive learning (RIL) algorithm is developed to maximize the data requester's cumulative utility by dynamically adjusting incentive levels according to workers' responses to payments. (3) We prove that our Bayesian inference algorithm and RIL algorithm are incentive compatible (IC) at each step and in the long run, respectively. (4) Experiments are conducted to test our mechanism, which shows that our mechanism performs consistently well under different worker models. Meanwhile, compared with the state-of-the-art peer prediction solutions, our Bayesian inference aided mechanism can improve the robustness and lower the variances of payments.

## 2  Problem Formulation

This paper considers the following data acquisition problem via crowdsourcing: at each discrete time step $t = 1, 2, ...$, a data requester assigns $M$ tasks with binary answer space $\{-1, +1\}$ to $N \geq 3$ candidate workers to label. Workers receive payments for submitting a label for each task. We use $L_i^t(j)$ to denote the label worker $i$ generates for task $j$ at time $t$. For simplicity of computation, we reserve $L_i^t(j) = 0$ if $j$ is not assigned to $i$. Furthermore, we use $\mathcal{L}$ and $\boldsymbol{L}$ to denote the set of ground-truth labels and the set of all collected labels respectively.

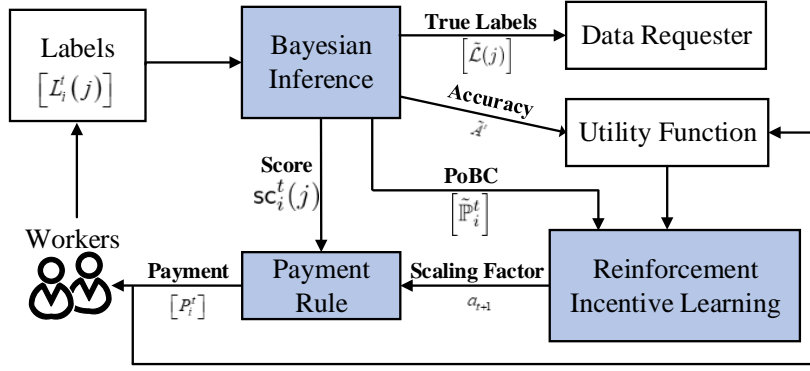

Figure 1: Overview of our incentive mechanism.

The generated label $L_i^t(j)$ depends both on the latent ground-truth $\mathcal{L}(j)$ and worker $i$'s strategy, which is mainly determined by two factors: exerted effort level (high or low) and reporting strategy (truthful or deceitful). Accommodating the notation commonly used in reinforcement learning, we also refer worker $i$'s strategy as his/her internal *state*. At any given time, workers at their will adopt an arbitrary combination of effort level and reporting strategy. Specifically, we define $\mathsf{eft}_i^t \in [0,1]$ and $\mathsf{rpt}_i^t \in [0,1]$ as worker $i$'s probability of exerting high efforts and reporting truthfully for task $j$, respectively. Furthermore, we use $\mathbb{P}_{i,H}$ and $\mathbb{P}_{i,L}$ to denote worker $i$'s probability of observing the true label when exerting high and low efforts, respectively. Correspondingly, we denote worker $i$'s cost of exerting high and low efforts by $c_{i,H}$ and $c_{i,L}$, respectively. For the simplicity of analysis, we assume that $\mathbb{P}_{i,H} > \mathbb{P}_{i,L} = 0.5$ and $c_{i,H} > c_{i,L} = 0$. All the above parameters and workers' actions stay unknown to our mechanism. In other words, we regard workers as black-boxes, which distinguishes our mechanism from the existing peer prediction mechanisms.

Worker $i$'s probability of being correct (PoBC) at time $t$ for any given task is given as

$$\mathbb{P}_i^t = \mathsf{rpt}_i^t \cdot \mathsf{eft}_i^t \cdot \mathbb{P}_{i,H} + (1 - \mathsf{rpt}_i^t) \cdot \mathsf{eft}_i^t \cdot (1 - \mathbb{P}_{i,H}) + \\ \mathsf{rpt}_i^t \cdot (1 - \mathsf{eft}_i^t) \cdot \mathbb{P}_{i,L} + (1 - \mathsf{rpt}_i^t) \cdot (1 - \mathsf{eft}_i^t) \cdot (1 - \mathbb{P}_{i,L}) \tag{1}$$

Suppose we assign $m_i^t \le M$ tasks to worker $i$ at step $t$. Then, a risk-neutral worker's utility satisfies:

$$u_i^t = \sum\nolimits_{j=1}^{M} P_i^t(j) - m_i^t \cdot c_{i,H} \cdot \mathsf{eft}_i^t \tag{2}$$

where $P_i^t$ denotes our payment to worker $i$ for task $j$ at time $t$ (see Section 3 for more details).

At the beginning of each step, the data requester and workers agree to a certain rule of payment, which is not changed until the next time step. The workers are self-interested and may choose their strategies in labeling and reporting according to the expected utility he/she can get. After collecting the generated labels, the data requester infers the true labels $\tilde{L}^t(j)$ by running a certain inference algorithm. The aggregated label accuracy $A^t$ and the data requester's utility $r_t$ are defined as follows:

$$A^t = \frac{1}{M} \sum\nolimits_{j=1}^{M} \mathbb{1}\left[\tilde{L}^t(j) = \mathcal{L}(j)\right] \ , \ r_t = F(A^t) - \eta \sum\nolimits_{i=1}^{N} \sum\nolimits_{j=1}^{M} P_i^t(j) \tag{3}$$

where $F(\cdot)$ is a non-decreasing monotonic function mapping accuracy to utility and $\eta > 0$ is a tunable parameter balancing label quality and costs.

# 3 Inference-Aided Reinforcement Mechanism for Crowdsourcing

Our mechanism mainly consists of three components: the payment rule, Bayesian inference and reinforcement incentive learning (RIL); see Figure 1 for an overview, where estimated values are denoted with tildes. The payment rule computes the payment to worker $i$ for his/her label on task $j$

$$P_i^t(j) = a_t \cdot [\mathsf{sc}_i^t(j) - 0.5] + b \tag{4}$$

where $a_t \in \mathcal{A}$ denotes the scaling factor, determined by RIL at the beginning of every step $t$ and shared by all workers. $\mathsf{sc}_i^t(j)$ denotes worker $i$'s score on task $j$, which will be computed by the Bayesian inference algorithm. $b \geq 0$ is a constant representing the fixed base payment. The Bayesian inference algorithm is also responsible for estimating the true labels, workers' PoBCs and the aggregated label accuracy at each time step, preparing the necessary inputs to RIL. Based on these estimates, RIL seeks to maximize the cumulative utility of the data requester by optimally balancing the utility (accuracy in labels) and the payments.

## 3.1 Bayesian Inference

For the simplicity of notation, we omit the superscript $t$ in this subsection. The motivation for designing our own Bayesian inference algorithm is as follows. We ran several preliminary experiments using popular inference algorithms, for example, EM [3, 16, 22] and Variational Inference [11, 1]). Our empirical studies reveal that those methods tend to heavily bias towards over-estimating the aggregated label accuracy when the quality of labels is low.[1] This leads to biased estimation of the data requester's utility $r_t$ (as it cannot be observed directly), and this estimated utility is used as the reward signal in RIL, which will be detailed later. Since the reward signal plays the core role in guiding the reinforcement learning process, the heavy bias will severely mislead our mechanism.

To reduce the estimation bias, we develop a Bayesian inference algorithm by introducing soft Dirichlet priors to both the distribution of true labels $\boldsymbol{\tau} = [\tau_{-1}, \tau_{+1}] \sim \mathrm{Dir}(\beta_{-1}, \beta_{+1})$, where $\tau_{-1}$ and $\tau_{+1}$ denote that of label $-1$ and $+1$, respectively, and workers' PoBCs $[\mathbb{P}_i, 1 - \mathbb{P}_i] \sim \mathrm{Dir}(\alpha_1, \alpha_2)$. Then, we derive the conditional distribution of true labels given collected labels as (see Appendix A) $\mathbb{P}(\mathcal{L}|\boldsymbol{L}) = \mathbb{P}(\boldsymbol{L}, \mathcal{L})/\mathbb{P}(\boldsymbol{L}) \propto B(\hat{\boldsymbol{\beta}}) \cdot \prod_{i=1}^N B(\hat{\boldsymbol{\alpha}}_i)$, where $B(x, y) = (x-1)!(y-1)!/(x+y-1)!$ denotes the beta function, $\hat{\boldsymbol{\alpha}} = [\hat{\alpha}_1, \hat{\alpha}_2], \hat{\boldsymbol{\beta}} = [\hat{\beta}_{-1}, \hat{\beta}_{+1}], \hat{\alpha}_{i1} = \sum_{j=1}^M \sum_{k \in \{-1, +1\}} \delta_{ijk} \xi_{jk} + 2\alpha_1 - 1$, $\hat{\alpha}_{i2} = \sum_{j=1}^M \sum_{k \in \{-1, +1\}} \delta_{ij(-k)} \xi_{jk} + 2\alpha_2 - 1$ and $\hat{\beta}_k = \sum_{j=1}^M \xi_{jk} + 2\beta_k - 1$, where $\delta_{ijk} = \mathbb{1}(L_i(j) = k)$ and $\xi_{jk} = \mathbb{1}(\mathcal{L}(j) = k)$.

Note that it is generally hard to derive an explicit formula for the posterior distribution of a specific task $j$'s ground-truth from the conditional distribution $\mathbb{P}(\mathcal{L}|\boldsymbol{L})$. We thus resort to Gibbs sampling for the inference. More specifically, according to Bayes' theorem, we know that the conditional distribution of task $j$'s ground-truth $\mathcal{L}(j)$ satisfies $\mathbb{P}[\mathcal{L}(j)|\boldsymbol{L}, \mathcal{L}(-j)] \propto \mathbb{P}(\mathcal{L}|\boldsymbol{L})$, where $-j$ denotes all tasks excluding $j$. Leveraging this, we gen-

---
**Algorithm 1** Gibbs sampling for crowdsourcing
---
1: **Input:** the collected labels $\boldsymbol{L}$, the number of samples $W$
2: **Output:** the sample sequence $\mathcal{S}$
3: $\mathcal{S} \leftarrow \emptyset$, Initialize $\mathcal{L}$ with the uniform distribution
4: **for** $s = 1$ **to** $W$ **do**
5:     **for** $j = 1$ **to** $M$ **do**
6:         $\mathcal{L}(j) \leftarrow 1$ and compute $x_1 = B(\hat{\boldsymbol{\beta}}) \prod_{i=1}^N B(\hat{\boldsymbol{\alpha}}_i)$
7:         $\mathcal{L}(j) \leftarrow 2$ and compute $x_2 = B(\hat{\boldsymbol{\beta}}) \prod_{i=1}^N B(\hat{\boldsymbol{\alpha}}_i)$
8:         $\mathcal{L}(j) \leftarrow$ Sample $\{1, 2\}$ with $P(1) = x_1/(x_1 + x_2)$
9:     Append $\tilde{\mathcal{L}}$ to the sample sequence $\mathcal{S}$
---

erate samples of the true label vector $\mathcal{L}$ following Algorithm 1. At each step of the sampling procedure (lines 6-7), Algorithm 1 first computes $\mathbb{P}[\mathcal{L}(j)|\boldsymbol{L}, \mathcal{L}(-j)]$ and then generates a new sample of $\mathcal{L}(j)$ to replace the old one in $\tilde{\mathcal{L}}$. After traversing through all tasks, Algorithm 1 generates a new sample of the true label vector $\mathcal{L}$. Repeating this process for $W$ times, we get $W$ samples, which is recorded in $\mathcal{S}$. Here, we write the $s$-th sample as $\tilde{\mathcal{L}}^{(s)}$. Since Gibbs sampling requires a burn-in process, we discard the first $W_0$ samples and calcualte worker $i$'s score on task $j$ and PoBC as

$$\mathsf{sc}_i^t(j) = \sum_{s=W_0}^W \frac{\mathbb{1}\left[\tilde{\mathcal{L}}^{(s)}(j) = L_i(j)\right]}{W - W_0}, \tilde{\mathbb{P}}_i = \frac{\sum\limits_{s=W_0}^W \left[2\alpha_1 - 1 + \sum\limits_{j=1}^M \mathbb{1}(\tilde{\mathcal{L}}^{(s)}(j) = L_i(j))\right]}{(W - W_0) \cdot (2\alpha_1 + 2\alpha_2 - 2 + m_i)}. \quad (5)$$

Similarly, we can obtain the estimates of the true label distribution $\boldsymbol{\tau}$ and then derive the log-ratio of task $j$, $\sigma_j = \log(\mathbb{P}[\mathcal{L}(j) = -1]/\mathbb{P}[\mathcal{L}(j) = +1])$. Furthermore, we decide the true label estimate $\tilde{\mathcal{L}}(j)$ as $-1$ if $\tilde{\sigma}_j > 0$ and as $+1$ if $\tilde{\sigma}_j < 0$. Correspondingly, the label accuracy $A$ is estimated as

$$\tilde{A} = \mathbb{E}(A) = M^{-1} \sum\nolimits_{j=1}^M e^{|\tilde{\sigma}_j|} \left(1 + e^{|\tilde{\sigma}_j|}\right)^{-1}. \quad (6)$$

In our Bayesian inference algorithm, workers' scores, PoBCs and the true label distribution are all estimated by comparing the true label samples with the collected labels. Thus, t To prove the convergence of our algorithm, we need to bound the ratio of wrong samples. We introduce $n$ and $m$ to denote the number of tasks of which the true label sample in Eqn. (5) is correct ($\tilde{\mathcal{L}}^{(s)}(j) = \mathcal{L}(j)$) and wrong ($\tilde{\mathcal{L}}^{(s)}(j) \neq \mathcal{L}(j)$) in the $s$-th sample, respectively. Formally, we have:

**Lemma 1.** *Let* $\bar{\mathbb{P}} = 1 - \mathbb{P}$, $\hat{\mathbb{P}} = \max\{\mathbb{P}, \bar{\mathbb{P}}\}$ *and* $\mathbb{P}_0 = \tau_{-1}$. *When* $M \gg 1$,

$$\mathbb{E}[m/M] \lesssim (1 + e^\delta)^{-1}(\varepsilon + e^\delta)(1 + \varepsilon)^{M-1} \ , \ \mathbb{E}[m/M]^2 \lesssim (1 + e^\delta)^{-1}(\varepsilon^2 + e^\delta)(1 + \varepsilon)^{M-2} \quad (7)$$

*where* $\varepsilon^{-1} = \prod_{i=0}^{N}(2\hat{\mathbb{P}}_i)^2$, $\delta = O[\Delta \cdot \log(M)]$ *and* $\Delta = \sum_{i=1}^{N}[1(\mathbb{P}_i < 0.5) - 1(\mathbb{P}_i > 0.5)]$.

The proof is in Appendix B. Our main idea is to introduce a set of counts for the collected labels and then calculate $\mathbb{E}[m/M]$ and $\mathbb{E}[m/M]^2$ based on the distribution of these counts. Using Lemma 1, the convergence of our Bayesian inference algorithm states as follows:

**Theorem 1** (Convergence). *When* $M \gg 1$ *and* $\prod_{i=0}^{N}(2\hat{\mathbb{P}}_i)^2 \geq M$, *if most of workers report truthfully (i.e.* $\Delta < 0$), *with probability at least* $1 - \delta \in (0, 1)$, $|\tilde{\mathbb{P}}_i - \mathbb{P}_i| \leq O(1/\sqrt{\delta M})$ *holds for any worker* $i$'s *PoBC estimate* $\tilde{\mathbb{P}}_i$ *as well as the true label distribution estimate (* $\tilde{\tau}_{-1} = \tilde{\mathbb{P}}_0$ *).*

The convergence of $\tilde{\mathbb{P}}_i$ and $\tilde{\tau}$ can naturally lead to the convergence of $\tilde{\sigma}_j$ and $\tilde{A}$ because the latter estimates are fully computed based on the former ones. All these convergence guarantees enable us to use the estimates computed by Bayesian inference to construct the state and reward signal in our reinforcement learning algorithm RIL. In addition, we summarize the time and space complexity of our Bayesian inference algorithm as follows.

**Theorem 2** (Complexity). *The time and space complexity of our Bayesian inference algorithm are* $O(WMN)$ *and* $O(WM)$, *respectively.*

The space complexity is obvious, as the two loops in Algorithm 1 generate $WM$ samples in total. About the time complexity, the additional term $N$ results from the calculation of $x_1$ and $x_2$. Since our Bayesian inference algorithm only considers one variable ($\mathcal{L}$), the convergence of sampling is sufficiently fast. In practice, the number of samples $W$ does not need to be too big. We set it to be 1000 in our experiments. Considering neither the number of tasks nor workers is very large, the computational efficiency of our Bayesian inference should be acceptable.

### 3.2 Reinforcement Incentive Learning

In this subsection, we formally introduce our reinforcement incentive learning (RIL) algorithm, which adjusts the scaling factor $a_t$ to maximize the data requesters' utility accumulated in the long run. To fully understand the technical background, readers are expected to be familiar with Q-value and function approximation. For readers with limited knowledge, we kindly refer them to Appendix D, where we provide background on these concepts. With transformation, our problem can be perfectly modeled as a Markov Decision Process. To be more specific, our mechanism is the agent and it interacts with workers (i.e. the environment); scaling factor $a_t$ is the action; the utility of the data requester $r_t$ defined in Eqn. (3) is the reward. Workers' reporting strategies are the state. After receiving payments, workers may change their strategies to, for example, increase their utilities at the next step. How workers change their strategies forms the state transition kernel.

On the other hand, the reward $r_t$ defined in Eqn. (3) cannot be directly used because the true accuracy $A^t$ cannot be observed. Thus, we use the estimated accuracy $\tilde{A}$ calculated by Eqn. (6) instead to approximate $r_t$ as in Eqn. (8). Furthermore, to achieve better generalization across different states, it is a common approach to learn a feature-based state representation $\phi(s)$ [14, 9]. Recall that the data requester's implicit utility at time $t$ only depends on the aggregated PoBC averaged across all workers. Such observation already points out to a representation design with good generalization, namely $\phi(s_t) = \sum_{i=1}^{N} \mathbb{P}_i^t / N$. Further recall that, when deciding the current scaling factor $a_t$, the data requester does not observe the latest workers' PoBCs and thus cannot directly estimate the current $\phi(s_t)$. Due to this one-step delay, we have to build our state representation using the previous observation. Since most workers would only change their internal states after receiving a new incentive, there exists some imperfect mapping function $\phi(s_t) \approx f(\phi(s_{t-1}), a_{t-1})$. Utilizing this implicit function, we introduce the augmented state representation in RIL as $\hat{s}_t$ in Eqn. (8).

$$r_t \approx F(\tilde{A}^t) - \eta \sum_{i=1}^{N} P_i^t \ , \ \hat{s}_t = \langle \phi(s_{t-1}), a_{t-1} \rangle. \quad (8)$$

Since neither $r_t$ nor $s_t$ can be perfectly inferred, it would not be a surprise to observe some noise that cannot be directly learned in our Q-function.

For most crowdsourcing problems the number of tasks $M$ is large, so we can leverage the central limit theorem to justify our modeling of the noise using a Gaussian process. To be more specific, we calculate the temporal difference (TD) error as

$$r_t \approx Q^\pi(\hat{s}_t, a_t) - \gamma \mathbb{E}_\pi Q^\pi(\hat{s}_{t+1}, a_{t+1}) + \epsilon_t \tag{9}$$

where the noise $\epsilon_t$ follows a Gaussian process, and $\pi = \mathbb{P}(a|\hat{s})$ denotes the current policy. By doing so, we gain two benefits. First, the approximation greatly simplifies the derivation of the update equation for the Q-function. Secondly, as shown in our empirical results later, this kind of approximation is robust against different worker models. Besides, following [6] we approximate Q-function as

---

**Algorithm 2** Reinforcement Incentive Learning (RIL)

1: **for** each episode **do**
2:    **for** each step in the episode **do**
3:       Decide the scaling factor as ($\epsilon$-greedy method)
$$a_t = \begin{cases} \arg\max_{a \in \mathcal{A}} Q(\hat{s}_t, a) & \text{Probability } 1 - \epsilon \\ \text{Random } a \in \mathcal{A} & \text{Probability } \epsilon \end{cases}$$
4:       Assign tasks and collect labels from the workers
5:       Run Bayesian inference to get $\hat{s}_{t+1}$ and $r_t$
6:       Use $(\hat{s}_t, a_t, r_t)$ to update $\boldsymbol{K}$, $\boldsymbol{H}$ and $\boldsymbol{r}$ in Eqn. (10)

---

$Q^\pi(\hat{s}_{t+1}, a_{t+1}) \approx \mathbb{E}_\pi Q^\pi(\hat{s}_{t+1}, a_{t+1}) + \epsilon_\pi$, where $\epsilon_\pi$ also follows a Gaussian process.

Under the Gaussian process approximation, all the observed rewards and the corresponding $Q$ values up to the current step $t$ form a system of equations, and it can be written as $\boldsymbol{r} = \boldsymbol{HQ} + \boldsymbol{N}$, where $\boldsymbol{r}$, $\boldsymbol{Q}$ and $\boldsymbol{N}$ denote the collection of rewards, $Q$ values, and residuals. Following Gaussian process's assumption for residuals, $\boldsymbol{N} \sim \mathcal{N}(\boldsymbol{0}, \boldsymbol{\sigma}^2)$, where $\boldsymbol{\sigma}^2 = \text{diag}(\sigma^2, \ldots, \sigma^2)$. The matrix $\boldsymbol{H}$ satisfies $\boldsymbol{H}(k, k) = 1$ and $\boldsymbol{H}(k, k+1) = -\gamma$ for $k = 1, \ldots, t$. Then, by using the online Gaussian process regression algorithm [5], we effectively learn the Q-function as

$$Q(\hat{s}, a) = \boldsymbol{k}(\hat{s}, a)^{\mathrm{T}} (\boldsymbol{K} + \boldsymbol{\sigma}^2)^{-1} \boldsymbol{H}^{-1} \boldsymbol{r} \tag{10}$$

where $\boldsymbol{k}(\hat{s}, a) = [k((\hat{s}, a), (\hat{s}_1, a_1)), \ldots, k((\hat{s}, a), (s_t, a_t))]^{\mathrm{T}}$ and $\boldsymbol{K} = [\boldsymbol{k}(\hat{s}_1, a_1), \ldots, \boldsymbol{k}(\hat{s}_t, a_t)]$. Here, we use $k(\cdot, \cdot)$ to denote the Gaussian kernel. Finally, we employ the classic $\epsilon$-greedy method to decide $a_t$ based on the learned Q-function. To summarize, we provide a formal description about RIL in Algorithm 2. Note that, when updating $\boldsymbol{K}$, $\boldsymbol{H}$ and $\boldsymbol{r}$ in Line 6, we employ the sparse approximation proposed in [6] to discard some data so that the size of these matrices does not increase infinitely.

## 4 Theoretical Analysis on Incentive Compatibility

In this section, we prove the incentive compatibility of our Bayesian inference and reinforcement learning algorithms. Our main results are as follows:

**Theorem 3** (One Step IC)**.** *At any time step $t$, when $M \gg 1$, $\prod_{i=1}^N (2\mathbb{P}_{i,H})^2 \geq M$, $a_t > \max_i c_{i,H}/(\mathbb{P}_{i,H} - 0.5)$, reporting truthfully and exerting high efforts is the utility-maximizing strategy for any worker $i$ at equilibrium (if other workers all follow this strategy).*

*Proof.* In Appendix E, we prove that when $a_t > c_{i,H}/(\mathbb{P}_{i,H} - 0.5)$, if $\tilde{\mathbb{P}}_i^t \approx \mathbb{P}_i^t$, any worker $i$'s utility-maximizing strategy would be reporting truthfully and exerting high efforts. Since Theorem 1 has provided the convergence guarantee, we can conclude Theorem 2. $\square$

**Theorem 4** (Long Term IC)**.** *Suppose the conditions in Theorem 3 are satisfied and the learned Q-function approaches the real $Q^\pi(\hat{s}, a)$. When the following equation holds for $i = 1, \ldots, N$,*

$$\eta M \sum_{x \neq i} \mathbb{P}_{x,H} \cdot G_{\mathcal{A}} > \frac{F(1) - F(1 - \psi_i)}{1 - \gamma} \; , \; \psi_i = \left( \frac{\tau_{-1}}{\tau_{+1}} + \frac{\tau_{+1}}{\tau_{-1}} \right) \prod_{x \neq i} \sqrt{4 \mathbb{P}_{x,H}(1 - \mathbb{P}_{x,H})} \tag{11}$$

*always reporting truthfully and exerting high efforts is the utility-maximizing strategy for any worker $i$ in the long term if other workers all follow this strategy. Here, $G_{\mathcal{A}} = \min_{a,b \in \mathcal{A}, a \neq b} |a - b|$ denotes the minimal gap between two available values of the scaling factor.*

In order to induce RIL to change actions, worker $i$ must let RIL learn a wrong $Q$-function. Thus, our main idea of proof is to derive the upper bounds of the effects of worker $i$'s reports on the $Q$-function. Besides, Theorem 3 points that, to design robust reinforcement learning algorithms against the manipulation of strategical agents, we should leave a certain level of gaps between actions. This observation may be of independent interests to reinforcement learning researchers.

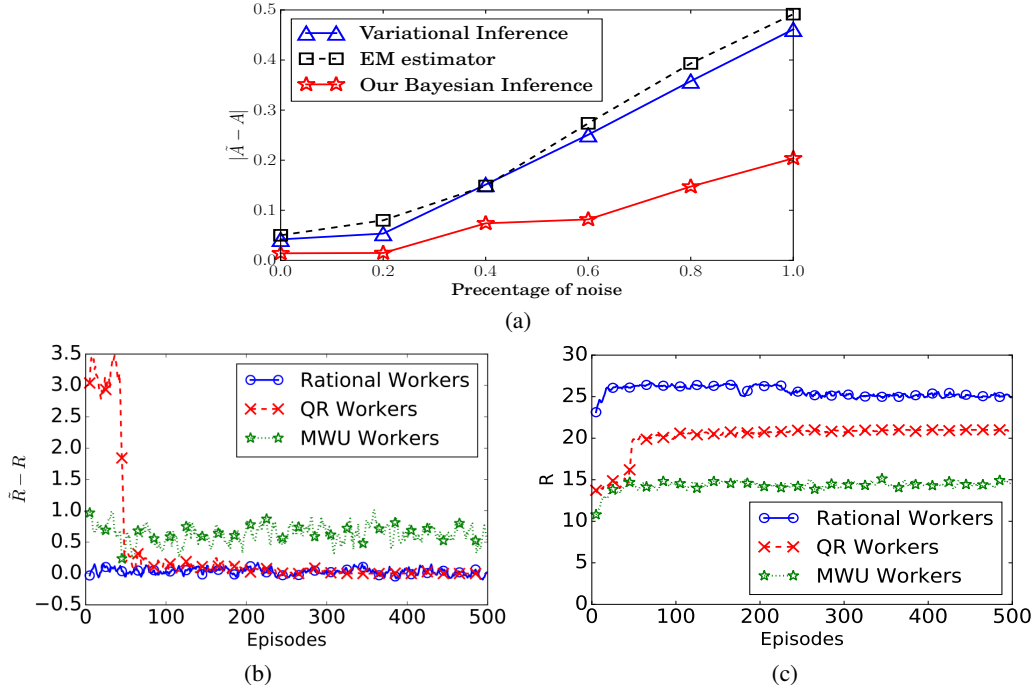

Figure 2: Empirical analysis on Bayesian Inference (a) and RIL (b-c). To be more specific, (a) compares the inference bias (i.e. the difference from the inferred label accuracy to the real one) of our Bayesian inference algorithm with that of EM and variational inference, averaged over 100 runs. (b) draws the gap between the estimation of the data requester's cumulative utility and the real one, smoothed over 5 episodes. (c) shows the learning curve of our mechanism, smoothed over 5 episodes.

## 5 Empirical Experiments

In this section, we empirically investigate the competitiveness of our solution. To be more specific, we first show our proposed Bayesian inference algorithm produces more accurate estimates for the aggregated label accuracy than the existing inference methods. Encouraged by the above results, we then move on to evaluate RIL on three popular worker models, each representing one classic way of modeling how human agent makes decisions. Note for this part, our experiments are conducted on the real-world data. Via these experiments, we demonstrate that our RIL algorithm consistently manages to learn a good incentive policy, under above three rationality models of the workers. Lastly, we show as a bonus benefit of our mechanism that, leveraging Bayesian inference to fully exploit the information contained in the collected labels leads to more robust and lower-variance payments at each step.

### 5.1 Empirical Analysis on Bayesian Inference

The aggregated label accuracy estimated from our Bayesian inference algorithm serves as a major component of the state representation and reward function to RIL, and thus critically affects the performance of our mechanism. Due to this, we choose to first investigate the bias of our Bayesian inference algorithm. To be more specific, we compare our Bayesian inference algorithm with two popular inference algorithms in crowdsourcing, that is, the EM estimator [16] and the variational inference estimator [11]. We utilize the RTE dataset, where workers need to check whether a hypothesis sentence can be inferred from the provided sentence [20]. In order to simulate strategic behaviors of workers, we mix these data with random noise by replacing a part of real-world labels with uniformly generated ones (low quality labels).

As Figure 2a suggests, compared to EM and variational inference, our proposed Bayesian inference algorithm can significantly lower the bias of the estimates of the aggregated label accuracy, and the

Table 1: Performance comparison under three worker models. Data requester's cumulative utility normalized over the number of tasks. Standard deviation reported in parenthesis.

| METHOD | RATIONAL | QR | MWU |
|---|---|---|---|
| FIXED OPTIMAL | 27.584 (.253) | 21.004 (.012) | 11.723 (.514) |
| HEURISTIC OPTIMAL | 27.643 (.174) | 21.006 (.001) | 12.304 (.515) |
| ADAPTIVE OPTIMAL | 27.835 (.209) | 21.314 (.011) | 17.511 (.427) |
| RIL | 27.184 (.336) | 21.016 (.018) | 15.726 (.416) |

advantage of our Bayesian inference algorithm is mostly in the high-noise regime. This is because both variational inference and EM maintain the updates on the posterior distribution via the estimates of workers' PoBCs. When the noise is very high, inaccurate PoBCs inevitably cause the posterior distribution to be biased. In our Bayesian inference algorithm, we introduce a soft Dirichlet prior for true labels, acting as a regularization term, to alleviate this bias.

In fact, we cannot use the estimates from either the EM or the variational inference as an alternative for the reward signal because the biases of their estimates are much higher when the noise level is high; for instance, this bias can reach $0.45$ while the range of the label accuracy is between $[0.5, 1.0]$. This set of experiments justifies our motivation to develop our own inference algorithm and reinforces our claim that our inference algorithm could provide fundamentals for the further development of potential learning algorithms for crowdsourcing.

## 5.2 Empirical Analysis on RIL

We move on to investigate whether RIL consistently learns a good policy, which maximizes the data requester's cumulative utility $R = \sum_t r_t$. For all the experiments in this subsection, we set the environment parameters as follows: $N = 10$, $\mathbb{P}_H = 0.9$, $b = 0$, $c_H = 0.02$; the set of the scaling factors is $\mathcal{A} = \{0.1, 1.0, 5.0, 10\}$; $F(A) = A^{10}$ and $\eta = 0.001$ as in the utility function (Eqn. (3)); the number of time steps for an episode is set to be $28$. Meanwhile, for the adjustable parameters in our mechanism, we set the number of tasks at each step $M = 100$ and the exploration rate for RIL $\epsilon = 0.2$. We report the averaged results over 5 runs to reduce the effect of outliers. To demonstrate our algorithm's general applicability, we test it under three different worker models, each representing a popular way for modeling how human agents make decisions. We provide a simple description of them as follows, whereas the detailed version is deferred to Appendix H. (i) *Rational* workers alway take the utility-maximizing strategies. (ii) *QR* workers [13] follow strategies corresponding to an utility-dependent distribution (which is pre-determined). This model has been used to study agents with bounded rationality. (iii) *MWU* workers [10] update their strategies according to the celebrated multiplicative weights update algorithm. This model has been used to study adaptive learning agents.

Our first set of experiments is a continuation to the last subsection. To be more specific, we first focus on the estimation bias of the data requester's cumulative utility $R$. This value is used as the reward in RIL and is calculated from the estimates of the aggregated label accuracy. This set of experiments aim to investigate whether our RIL module successfully leverages the label accuracy estimates and picks up the right reward signal. As Figure 2b shows, the estimates only deviate from the real values in a very limited magnitude after a few episodes of learning, regardless of which worker model the experiments run with. The results further demonstrate that our RIL module observe reliable rewards. The next set of experiments is about how quickly RIL learns. As Figure 2c shows, under all three worker models, RIL manages to pick up and stick to a promising policy in less than 100 episodes. This observation also demonstrates the robustness of RIL under different environments.

Our last set of experiments in this subsection aim to evaluate the competitiveness of the policy learned by RIL. In Table 1, we use the policy learned after 500 episodes with exploration rate turned off (i.e. $\epsilon = 0$) and compare it with three benchmarks constructed by ourselves. To create the first one, Fixed Optimal, we try all 4 possible fixed value for the scaling factor and report the highest cumulative reward realized by either of them. To create the second one, Heuristic Optimal, we divide the value region of $\tilde{A}^t$ into five regions: $[0, 0.6)$, $[0.6, 0.7)$, $[0.7, 0.8)$, $[0.8, 0.9)$ and $[0.9, 1.0]$. For each region, we select a fixed value for the scaling factor $a_t$. We traverse all $4^5 = 1024$ possible combinations to decide the optimal heuristic strategy. To create the third one, Adaptive Optimal, we change the scaling factor every 4 steps and report the highest cumulative reward via traversing all $4^7 = 16384$ possible

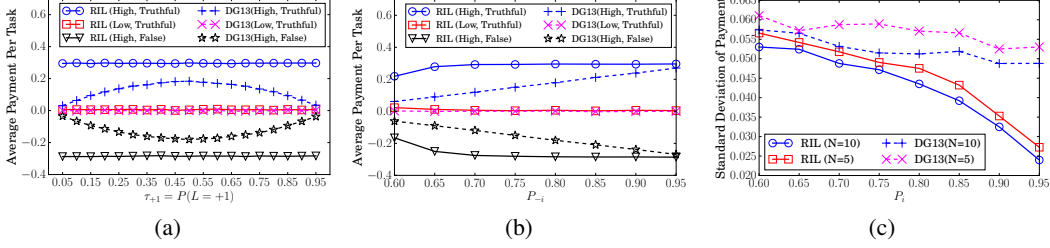

Figure 3: Empirical analysis on our Bayesian inference algorithm, averaged over 1000 runs. (a) Average payment per task given true label's distribution. (b) Average payment per task given PoBCs of workers excluding $i$. (c) The standard deviation of the payment given worker $i$'s PoBC.

configurations. This benchmark is infeasible to be reproduced in real-world practice, once the number of steps becomes large. Yet it is very close to the global optimal in the sequential setting. As Table 1 demonstrates, the two benchmarks plus RIL all achieve a similar performance tested under rational and QR workers. This is because these two kinds of workers have a fixed pattern in responding to incentives and thus the optimal policy would be a fixed scaling factor throughout the whole episode. On contrast, MWU workers adaptively learn utility-maximizing strategies gradually, and the learning process is affected by the incentives. Under this worker environment, RIL managers to achieve an average utility score of 15.6, which is a significant improvement over fixed optimal and heuristic optimal (which achieve 11.7 and 12.3 respectively) considering the unrealistic global optimal is only around 18.5. Up to this point, with three sets of experiments, we demonstrate the competitiveness of RIL and its robustness under different work environments. Note that, when constructing the benchmarks, we also conduct experiments on DG13, the state-of-the-art peer prediction mechanism for binary labels [2], and get the same conclusion. For example, when DG13 and MWU workers are tested for Fixed Optimal and Heuristic Optimal, the cumulative utilities are 11.537(.397) and 11.908(0.210), respectively, which also shows a large gap with RIL.

### 5.3 Empirical Analysis on One Step Payments

In this subsection, we compare the one step payments provided by our mechanism with the payments calculated by DG13, the state-of-the-art peer prediction mechanism for binary labels [2]. We fix the scaling factor $a_t = 1$ and set $M = 100$, $N = 10$, $\mathbb{P}_H = 0.8$, $b = 0$ and $m_i^t = 90$. To set up the experiments, we generate task $j$'s true label $\mathcal{L}(j)$ following its distribution $\tau$ (to be specified) and worker $i$'s label for task $j$ based on $i$'s PoBC $\mathbb{P}_i$ and $\mathcal{L}(j)$. In Figure 3a, we let all workers excluding $i$ report truthfully and exert high efforts (i.e. $\mathbb{P}_{-i} = \mathbb{P}_H$), and increase $\tau_{+1}$ from 0.05 to 0.95. In Figure 3b, we let $\tau_{+1} = 0.5$, and increase other workers' PoBCs $\mathbb{P}_{-i}$ from 0.6 to 0.95. As both figures reveal, in our mechanism, the payment for worker $i$ almost only depends on his/her own strategy. On contrast, in DG13, the payments are clearly affected by the distribution of true labels and the strategies of other workers. In other words, our Bayesian inference is more robust to different environments. Furthermore, in Figure 3c, we present the standard deviation of the payment to worker $i$. We let $\tau_{+1} = 0.5$, $\mathbb{P}_{-i} = \mathbb{P}_H$ and increase $\mathbb{P}_i$ from 0.6 to 0.95. As shown in the figure, our method manages to achieve a noticeably smaller standard deviation compared to DG13. Note that, in Figure 3b, we implicitly assume that most of workers will at least not adversarially report false labels, which is widely-adopted in previous studies [11]. For workers' collusion attacks, we also have some defending tricks provided in Appendix F.

## 6 Conclusion

In this paper, we build an inference-aided reinforcement mechanism leveraging Bayesian inference and reinforcement learning techniques to learn the optimal policy to incentivize high-quality labels from crowdsourcing. Our mechanism is proved to be incentive compatible. Empirically, we show that our Bayesian inference algorithm can help improve the robustness and lower the variance of payments, which are favorable properties in practice. Meanwhile, our reinforcement incentive learning (RIL) algorithm ensures our mechanism to perform consistently well under different worker models.

## Acknowledgments

This work started when Zehong Hu was at the Rolls-Royce@NTU Corporate Lab with support from the National Research Foundation (NRF) Singapore under the Corp Lab@University Scheme. Yitao is partially supported by NSF grants #IIS-1657613, #IIS-1633857 and DARPA XAI grant #N66001-17-2-4032. Yang Liu acknowledges supports from NSF CCF #1718549. The authors also thank Anxiang Zeng from Alibaba Group for valuable discussions.

## Footnotes

[1]See Section 5.1 for detailed experiment results and analysis.

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
