[Supplementary Material · Supplementary.pdf]

# Appendix for "Inference Aided Reinforcement Learning for Incentive Mechanism Deisgn in Crowdsourcing"

**Zehong Hu**
Alibaba Group, Hangzhou, China
HUZE0004@e.ntu.edu.sg

**Yitao Liang**
University of California, Los Angeles
yliang@cs.ucla.edu

**Jie Zhang**
Nanyang Technological University
ZhangJ@ntu.edu.sg

**Zhao Li**
Alibaba Group, Hangzhou, China
lizhao.lz@alibaba-inc.com

**Yang Liu**
University of California, Santa Cruz/Harvard University
yangliu@ucsc.edu

## A Derivation of Posterior Distribution

It is not had to figure out the joint distribution of the collected labels $\boldsymbol{L}$ and the true labels $\mathcal{L}$

$$\mathbb{P}(\boldsymbol{L}, \mathcal{L}|\boldsymbol{\theta}, \boldsymbol{\tau}) = \prod_{j=1}^{M} \prod_{k \in \{-1, +1\}} \left\{ \tau_k \prod_{i=1}^{N} \mathbb{P}_i^{\delta_{ijk}} (1 - \mathbb{P}_i)^{\delta_{ij(-k)}} \right\}^{\xi_{jk}} \tag{12}$$

where $\boldsymbol{\theta} = [\mathbb{P}_1, \ldots, \mathbb{P}_N]$ and $\boldsymbol{\tau} = [\tau_{-1}, \tau_{+1}]$. $\tau_{-1}$ and $\tau_{+1}$ denote the distribution of true label $-1$ and $+1$, respectively. Besides, $\delta_{ijk} = \mathbb{1}(L_i(j) = k)$ and $\xi_{jk} = \mathbb{1}(\mathcal{L}(j) = k)$. Then, the joint distribution of $\boldsymbol{L}$, $\mathcal{L}$, $\boldsymbol{\theta}$ and $\boldsymbol{\tau}$

$$\mathbb{P}(\mathcal{L}, \boldsymbol{L}, \boldsymbol{p}, \boldsymbol{\tau}) = \mathbb{P}(\mathcal{L}, \boldsymbol{L}|\boldsymbol{p}, \boldsymbol{\tau}) \cdot \mathbb{P}(\boldsymbol{\theta}, \boldsymbol{\tau})$$

$$= \frac{1}{B(\boldsymbol{\beta})} \prod_{k \in \{-1, +1\}} \tau_k^{\hat{\beta}_k^* - 1} \cdot \prod_{i=1}^{N} \frac{1}{B(\boldsymbol{\alpha})} p_i^{\hat{\alpha}_{i1}^* - 1} (1 - p_i)^{\hat{\alpha}_{i2}^* - 1} \tag{13}$$

where $B(x, y) = (x - 1)!(y - 1)!/(x + y - 1)!$ denotes the beta function, and

$$\hat{\alpha}_{i1}^* = \sum_{j=1}^{M} \sum_{k=1}^{K} \delta_{ijk} \xi_{jk} + \alpha_1$$

$$\hat{\alpha}_{i2}^* = \sum_{j=1}^{M} \sum_{k=1}^{K} \delta_{ij(3-k)} \xi_{jk} + \alpha_2$$

$$\hat{\beta}_k^* = \sum_{j=1}^{M} \xi_{jk} + \beta_k.$$

In this case, we can conduct marginalization via integrating the joint distribution $\mathbb{P}(\mathcal{L}, \boldsymbol{L}, \boldsymbol{p}, \boldsymbol{\tau})$ over $\boldsymbol{\theta}$ and $\boldsymbol{\tau}$ as

$$P(\mathcal{L}, \boldsymbol{L}|\boldsymbol{\alpha}, \boldsymbol{\beta}) = \frac{B(\hat{\boldsymbol{\beta}})}{B(\boldsymbol{\beta})} \cdot \prod_{i=1}^{N} \frac{B(\hat{\boldsymbol{\alpha}}_i)}{[B(\boldsymbol{\alpha})]^2} \tag{14}$$

where $\hat{\boldsymbol{\alpha}}_i = [\hat{\alpha}_{i1}^* + \alpha_1 - 1, \hat{\alpha}_{i2}^* + \alpha_2 - 1]$ and $\hat{\boldsymbol{\beta}} = [\hat{\beta}_{-1}^* + \beta_{-1} - 1, \hat{\beta}_{+1}^* + \beta_{+1} - 1]$. Following Bayes' theorem, we can know that

$$P(\boldsymbol{L}|\mathcal{L}) = \frac{P(\mathcal{L}, \boldsymbol{L}|\boldsymbol{\alpha}, \boldsymbol{\beta})}{P(\mathcal{L}|\boldsymbol{\alpha}, \boldsymbol{\beta})} \propto B(\hat{\boldsymbol{\beta}}) \prod_{i=1}^{N} B(\hat{\boldsymbol{\alpha}}_i). \tag{15}$$

# B   Proof for Lemma 1

## B.1   Basic Lemmas

We firstly present some lemmas for our proof later.

**Lemma 2.** *If $x \sim \mathrm{Bin}(n, p)$, $\mathbb{E}t^x = (1 - p + tp)^n$ holds for any $t > 0$, where $\mathrm{Bin}(\cdot)$ is the binomial distribution.*

*Proof.*

$$t^x = e^{x \log t} = m_x(\log t) = \left(1 - p + pe^{\log t}\right)^n \tag{16}$$

where $m_x(\cdot)$ denotes the moment generating function. $\qquad \square$

**Lemma 3.** *For given $n, m \geq 0$, if $0 \leq p \leq 1$, we can have*

$$\sum_{x=0}^{n} \sum_{w=0}^{m} C_n^x C_m^w p^{x+w} (1-p)^{y+z} \times$$
$$B(x + z + 1 + t, y + w + 1) =$$
$$\int_0^1 [(2p-1)x + 1 - p]^n [(1-2p)x + p]^m x^t \mathrm{d}x$$

*Proof.* By the definition of the beta function [7],

$$B(x, y) = \int_0^{+\infty} u^{x-1}(1+u)^{-(x+y)} \mathrm{d}u \tag{17}$$

we can have

$$\sum_{x,w} C_n^x C_m^w p^{x+w} (1-p)^{y+z} B(x + z + 1 + t, y + w + 1)$$
$$= \int_0^{+\infty} \mathbb{E}u^x \cdot \mathbb{E}u^z \cdot u^t \cdot (1+u)^{-(n+m+2+t)} \mathrm{d}u \tag{18}$$

where we regard $x \sim \mathrm{Bin}(n, p)$ and $z \sim \mathrm{Bin}(m, 1-p)$. Thus, according to Lemma 2, we can obtain

$$\int_0^{+\infty} \mathbb{E}u^x \cdot \mathbb{E}u^z \cdot u^t \cdot (1+u)^{-(n+m+3)} \mathrm{d}u$$
$$= \int_0^{+\infty} \frac{[1 - p + up]^n \cdot [p + (1-p)u]^m \cdot u^t}{(1+u)^{n+m+2+t}} \mathrm{d}u. \tag{19}$$

For the integral operation, substituting $u$ with $v - 1$ at first and then $v$ with $(1-x)^{-1}$, we can conclude Lemma 3. $\qquad \square$

**Lemma 4.** $\sum_{n=0}^{N} C_N^n \cdot x^n = (1 + x)^N.$

**Lemma 5.** $\sum_{n=0}^{N} C_N^n \cdot n \cdot x^n = N \cdot x \cdot (1 + x)^{N-1}.$

**Lemma 6.** $\sum_{n=0}^{N} C_N^n \cdot n \cdot x^{N-n} = N \cdot (1 + x)^{N-1}.$

**Lemma 7.** $\sum_{n=0}^{N} C_N^n \cdot n^2 \cdot x^n = Nx(1 + Nx)(1 + x)^{N-2}.$

**Lemma 8.** $\sum_{n=0}^{N} C_N^n \cdot n^2 \cdot x^{N-n} = N(x + N)(1 + x)^{N-2}.$

**Lemma 9.** *If $0 < x < 1$, we can have*

$$\sum_{n=0}^{\lfloor N/2 \rfloor} C_N^n \cdot x^n \geq \left(1 - e^{-cN}\right) \cdot (1 + x)^N$$

$$\sum_{n=\lfloor N/2 \rfloor + 1}^{N} C_N^n \cdot x^{N-n} \geq \left(1 - e^{-cN}\right) \cdot (1 + x)^N.$$

*where $c = 0.5(1-x)^2(1+x)^{-2}$.*

*Proof.* To prove the lemmas above, we firstly define

$$F_t(x) = \sum_{n=0}^{N} C_N^n n^t x^n \tag{20}$$

Then, Lemma 4 can be obtained by expanding $(1+x)^N$. Lemma 5 can be proved as follows

$$F_1(x) = \sum_{n=0}^{N} C_N^n (n+1)x^n - (1+x)^N$$

$$\sum_{n=0}^{N} C_N^n (n+1)x^n = \frac{\mathrm{d}}{\mathrm{d}x}[xF_0(x)] \tag{21}$$

$$= Nx(1+x)^{N-1} + (1+x)^N.$$

Lemma 6 can be obtained as follows

$$\sum_{n=0}^{N} C_N^n n x^{N-n} = x^N \sum_{n=0}^{N} C_N^n n \left(\frac{1}{x}\right)^n$$

$$= x^N \cdot N \cdot \frac{1}{x} \cdot \left(1 + \frac{1}{x}\right)^{N-1}. \tag{22}$$

For Lemma 7, we can have

$$F_2(x) = \sum_{n=0}^{N} C_N^n (n+2)(n+1)x^n - 3F_1(x) - 2F_0(x)$$

$$= [x^2 F_0(x)]' - 3F_1(x) - 2F_0(x) \tag{23}$$

Thus, we can have

$$F_2(x) = Nx(1+Nx)(1+x)^{N-2} \tag{24}$$

which concludes Lemma 7. Then, Lemma 8 can be obtained by considering Eqn. (25).

$$\sum_{n=0}^{N} C_N^n n^2 x^{N-n} = x^N \sum_{n=0}^{N} C_N^n n^2 \left(\frac{1}{x}\right)^n. \tag{25}$$

For Lemma 9, we can have

$$\sum_{n=0}^{\lfloor N/2 \rfloor} C_N^n x^n = (1+x)^N \sum_{n=0}^{\lfloor N/2 \rfloor} C_N^n p^n (1-p)^{N-n} \tag{26}$$

where $p = x(1+x)^{-1}$. Let $X \sim \text{Bin}(N, p)$, we can have

$$\sum_{n=0}^{\lfloor N/2 \rfloor} C_N^n p^n (1-p)^{N-n} \geq 1 - P\left(X \geq N/2\right). \tag{27}$$

Since $x < 1$, $p < 0.5$ and $Np < N/2$. Considering Hoeffding's inequality, we can get

$$P\left(X \geq N/2\right) \leq \exp\left[-\frac{N(1-x)^2}{2(1+x)^2}\right] \tag{28}$$

which concludes the first inequality in Lemma 9. Similarly, for the second inequality, we can have

$$\sum_{n=K}^{N} C_N^n x^{N-n} = (1+x)^N \sum_{n=K}^{N} C_N^n (1-p)^n p^{N-n} \tag{29}$$

where $K = \lfloor N/2 \rfloor + 1$. Suppose $Y \sim \text{Bin}(N, 1-p)$, we can have

$$\sum_{n=K}^{N} C_N^n (1-p)^n p^{N-n} \geq 1 - P\left(Y \leq N/2\right). \tag{30}$$

Considering Hoeffding's inequality, we can also get

$$P\left(Y \leq N/2\right) \leq \exp\left[-\frac{N(1-x)^2}{2(1+x)^2}\right]$$ (31)

which concludes the second inequality in Lemma 9. □

**Lemma 10.** *For any $x, y \geq 0$, we can have*

$$(1+x)^y \leq e^{xy}.$$

*Proof.* Firstly, we can know $(1+x)^y = e^{y\log(1+x)}$. Let $f(x) = x - \log(x)$. Then, we can have $f(0) = 0$ and $f'(x) \geq 0$. Thus, $x \geq \log(1+x)$ and we can conclude Lemma 10 by taking this inequality into the equality. □

**Lemma 11.**

$$g(x) = \frac{e^x}{e^x + 1}$$

*is a concave function when $x \in [0, +\infty)$.*

*Proof.* $g'(x) = (2 + t(x))^{-1}$, where $t(x) = e^x + e^{-x}$. $t'(x) = e^x - e^{-x} \geq 0$ when $x \in [0, +\infty)$. Thus, $g'(x)$ is monotonically decreasing when $x \in [0, +\infty)$, which concludes Lemma 11. □

**Lemma 12.** *For $x \in (-\infty, +\infty)$,*

$$h(x) = \frac{1}{e^{|x|} + 1}$$

*satisfies*

$$h(x) < e^x \quad \text{and} \quad h(x) < e^{-x}.$$

*Proof.* When $x \geq 0$, we can have

$$h(x) < \frac{1}{e^x} = e^{-x} \leq e^x.$$ (32)

When $x \leq 0$, we can have

$$h(x) = \frac{e^x}{e^x + 1} < e^x \leq e^{-x}.$$ (33)

□

**Lemma 13.** *If $\lambda = p/(1-p)$ and $0.5 < p < 1$, then*

$$\sum_{n=\lfloor N/2 \rfloor}^{N} C_N^n \lambda^{m-n} p^n (1-p)^m \leq [4p(1-p)]^{N/2}$$

$$\sum_{n=0}^{\lfloor N/2 \rfloor} C_N^n \lambda^{n-m} p^n (1-p)^m \leq [4p(1-p)]^{N/2}$$

*where $m = N - n$.*

*Proof.* For the first inequality, we can have

$$\sum_{n=\lfloor N/2 \rfloor}^{N} C_N^n \lambda^{m-n} p^n (1-p)^m$$ (34)

$$= \sum_{n=\lfloor N/2 \rfloor}^{N} C_N^n p^m (1-p)^n \leq \sum_{m=0}^{\lfloor N/2 \rfloor} C_N^m p^m (1-p)^n$$

According to the inequality in [1], we can have

$$\sum_{m=0}^{\lfloor N/2 \rfloor} C_N^m p^m (1-p)^n \leq \exp(-ND)$$ (35)

where $D = -0.5\log(2p) - 0.5\log(2(p-1))$, which concludes the first inequality in Lemma 13.

For the second inequality, we can have

$$\sum_{n=0}^{\lfloor N/2 \rfloor} C_N^n \lambda^{n-m} p^n (1-p)^m$$

$$= \frac{1}{[p(1-p)]^N} \sum_{n=0}^{\lfloor N/2 \rfloor} C_N^n [p^3]^n [(1-p)^3]^m \tag{36}$$

$$= \frac{[p^3 + (1-p)^3]^N}{[p(1-p)]^N} \sum_{n=0}^{\lfloor N/2 \rfloor} C_N^n x^n (1-x)^m$$

where $x = p^3 / [p^3 + (1-p)^3]$. By using Eqn. (35), we can have

$$\sum_{n=0}^{\lfloor N/2 \rfloor} C_N^n \lambda^{n-m} p^n (1-p)^m$$

$$\leq \frac{[p^3 + (1-p)^3]^N}{[p(1-p)]^N} [x(1-x)]^{N/2} \tag{37}$$

$$= [4p(1-p)]^{N/2}$$

which concludes the second inequality of Lemma 13. ∎

## B.2 Main Proof

To prove Lemma 1, we need to analyze the posterior distribution of $\mathcal{L}$ which satisfies

$$\mathbb{P}(\mathcal{L}|\boldsymbol{L}) = B(\hat{\boldsymbol{\beta}}) \prod_{i=1}^N B(\hat{\boldsymbol{\alpha}}_i) / [C_p \cdot \mathbb{P}(\boldsymbol{L})] \tag{38}$$

where $C_p$ is the nomalization constant. This is because the samples are generated based on this distribution. However, both the numerator and denominator in Eqn. (38) are changing with $\boldsymbol{L}$, making the distribution difficult to analyze. Thus, we derive a proper approximation for the denominator of Eqn. (38) at first. Denote the labels generated by $N$ workers for task $j$ as vector $\boldsymbol{L}(j)$. The distribution of $\boldsymbol{L}(j)$ satisfies

$$\mathbb{P}_{\hat{\boldsymbol{\theta}}}[\boldsymbol{L}(j)] = \sum_{k \in \{-1,+1\}} \tau_k \prod_{i=1}^N \mathbb{P}_i^{\delta_{ijk}} (1 - \mathbb{P}_i)^{\delta_{ij(-k)}} \tag{39}$$

where $\hat{\boldsymbol{\theta}} = [\tau_{-1}, \mathbb{P}_1, \ldots, \mathbb{P}_N]$ denotes all the parameters and $\delta_{ijk} = \mathbb{1}(L_i(j) = k)$. Then, we can have

**Lemma 14.** *When $M \to \infty$,*

$$\mathbb{P}(\boldsymbol{L}) \to C_L(M) \cdot \prod_{\boldsymbol{L}(j)} \{\mathbb{P}_{\hat{\boldsymbol{\theta}}}[\boldsymbol{L}(j)]\}^{M \cdot \mathbb{P}_{\hat{\boldsymbol{\theta}}}[\boldsymbol{L}(j)]}$$

*where $C_L(M)$ denotes a constant that depends on $M$.*
*Proof.* Denote the prior distribution of $\boldsymbol{\theta}$ by $\pi$. Then,

$$P(\mathcal{L}|\boldsymbol{\alpha}, \boldsymbol{\beta}) = \prod_{j=1}^M P_{\boldsymbol{\theta}}(\boldsymbol{x}_j) \int e^{[-M \cdot d_{KL}]} \mathrm{d}\pi(\hat{\boldsymbol{\theta}}) \tag{40}$$

$$d_{KL} = \frac{1}{M} \sum_{j=1}^M \log \frac{P_{\boldsymbol{\theta}}(\boldsymbol{x}_j)}{P_{\hat{\boldsymbol{\theta}}}(\boldsymbol{x}_j)} \to \mathrm{KL}[P_{\boldsymbol{\theta}}(\boldsymbol{x}), P_{\hat{\boldsymbol{\theta}}}(\boldsymbol{x})] \tag{41}$$

where $\boldsymbol{x}_j$ denotes the labels generated for task $j$. The KL divergence $\mathrm{KL}[\cdot, \cdot]$, which denotes the expectation of the log-ratio between two probability distributions, is a constant for the given $\boldsymbol{\theta}$ and $\hat{\boldsymbol{\theta}}$. Thus, $\int e^{[-M \cdot d_{KL}]} \mathrm{d}\pi(\hat{\boldsymbol{\theta}}) = C_L(M)$. In addition, when $M \to \infty$, we can also have $\sum \mathbb{1}(\boldsymbol{x}_j = \boldsymbol{x}) \to M \cdot P_{\boldsymbol{\theta}}(\boldsymbol{x})$, which concludes Lemma 14. ∎

Then, we move our focus back to the samples. To quantify the effects of the collected labels, we introduce a set of variables to describe the real true labels and the collected labels. Among the $n$ tasks of which the posterior true label is correct,

- $x_0$ and $y_0$ denote the number of tasks of which the real true label is $-1$ and $+1$, respectively.
- $x_i$ and $y_i$ denote the number of tasks of which worker $i$'s label is correct and wrong, respectively.

Also, among the remaining $m = M - n$ tasks,

- $w_0$ and $z_0$ denote the number of tasks of which the real true label is $-1$ and $+1$, respectively.
- $w_i$ and $z_i$ denote the number of tasks of which worker $i$'s label is correct and wrong, respectively.

Thus, we can have $x_i + y_i = n$ and $w_i + z_i = m$. Besides, we use $\xi_i$ to denote the combination $(x_i, y_i, w_i, z_i)$.

To compute the expectation of $m/M$, we need to analyze the probability distribution of $m$. According to Eqn. (15), we can know that $\mathbb{P}(m)$ satisfies

$$\mathbb{P}(m) \approx \frac{C_M^m}{Z} \sum_{\xi_0, \dots, \xi_N} \prod_{i=0}^{N} \mathbb{P}(\xi_i|m) B(\hat{\boldsymbol{\beta}}) \prod_{i=1}^{N} B(\hat{\boldsymbol{\alpha}}_i) \tag{42}$$

where $Z = C_p C_L \prod_{\boldsymbol{x}} [P_{\boldsymbol{\theta}}(\boldsymbol{x})]^{M \cdot P_{\boldsymbol{\theta}}(\boldsymbol{x})}$ is independent of $\xi_i$ and $m$. Meanwhile, $\hat{\beta}_{-1} = x_0 + z_0 + 1$, $\hat{\beta}_{+1} = y_0 + w_0 + 1$, $\hat{\alpha}_{i1} = x_i + z_i + 2$ and $\hat{\alpha}_{i2} = x_i + z_i + 1$. When the $m$ tasks of which the posterior true label is wrong are given, we can know that $x_i \sim \text{Bin}(n, \mathbb{P}_i)$ and $w_i \sim \text{Bin}(m, \mathbb{P}_i)$, where $\text{Bin}(\cdot)$ denotes the binomial distribution. In addition, $x_i$ and $y_i$ are independent of $w_i$, $z_i$ and $\xi_{k \neq i}$. Also, $w_i$ and $z_i$ are independent of $x_i$ and $y_i$ and $\xi_{k \neq i}$. Thus, we can further obtain $\mathbb{P}(m) \approx \hat{Z}^{-1} \cdot C_M^m Y(m)$, where

$$Y(m) = e^{\log H(m, \mathbb{P}_0; M, 0) + \sum_{i=1}^{N} \log H(m, \mathbb{P}_i; M, 1)}$$
$$H(m, p; M, t) = \sum_{x=0}^{n} \sum_{w=0}^{m} 2^{M+1} C_n^x C_m^w \times \tag{43}$$
$$p^{x+w}(1-p)^{y+z} B(x + z + 1 + t, y + w + 1)$$

and $\hat{Z} = 2^{-(N+1)(M+1)} Z$. Considering $\sum_{m=1}^{M} \mathbb{P}(m) = 1$, we can know that $\hat{Z} \approx \sum_{m=1}^{M} C_M^m Y(m)$. Note that, we use $\mathbb{P}_0$ to denote the probability of true label 1, namely $\tau_1$.

**The biggest challenge of our proof** exists in analyzing function $H(m, p; M, t)$ which we put in the next subsection (Section C.3). Here, we directly use the obtained lower and upper bounds depicted in Lemmas 19 and 20 and can have

$$\begin{cases} e^{C - K_l m} \lesssim Y(m) \lesssim e^{C - K_u m} & 2m \leq M \\ e^{C + \delta - K_l n} \lesssim Y(m) \lesssim e^{C + \delta - K_u n} & 2m > M \end{cases} \tag{44}$$

where $C = H(0, \mathbb{P}_0; M, 0) + \sum_{i=1}^{N} H(0, \mathbb{P}_i; M, 1)$ and

$$K_l = \sum_{i=0}^{N} \log \hat{\lambda}_i \ , \ K_u = 2 \sum_{i=0}^{N} \log\left(2\hat{\mathbb{P}}_i\right)$$
$$\delta = \Delta \cdot \log(M) + \sum_{i=1}^{N} (-1)^{\mathbf{1}(\mathbb{P}_i > 0.5)} \phi(\hat{\mathbb{P}}_i)$$
$$\hat{\lambda}_i = \max\left\{\frac{\mathbb{P}_i}{\mathbb{P}_i + \frac{1}{M}}, \frac{\bar{\mathbb{P}}_i}{\bar{\mathbb{P}}_i + \frac{1}{M}}\right\} \ , \ \phi(p) = \log \frac{2\mathbb{P} - 1}{\mathbb{P}}$$
$$\Delta = \sum_{i=1}^{N} [\mathbf{1}(\mathbb{P}_i < 0.5) - \mathbf{1}(\mathbb{P}_i > 0.5)].$$

Here, $\bar{\mathbb{P}} = 1 - \mathbb{P}$, $\hat{\mathbb{P}} = \max\{\mathbb{P}, \bar{\mathbb{P}}\}$ and $\mathbb{P}_0 = \tau_{-1}$. Besides, we set a convention that $\phi(p) = 0$ when $p = 0.5$. Thereby, the expectations of $m$ and $m^2$ satisfy

$$\mathbb{E}[m] \lesssim \frac{\sum_{m=0}^{M} m e^{-K_u m} + \sum_{m=0}^{M} m e^{\delta - K_u n}}{\sum_{m=0}^{k} e^{-K_l m} + \sum_{m=k+1}^{M} e^{\delta - K_l n}} \tag{45}$$

$$\mathbb{E}[m^2] \lesssim \frac{\sum_{m=0}^{M} m^2 e^{-K_u m} + \sum_{m=0}^{M} m^2 e^{\delta - K_u n}}{\sum_{m=0}^{k} e^{-K_l m} + \sum_{m=k+1}^{M} e^{\delta - K_l n}} \tag{46}$$

where $k = \lfloor M/2 \rfloor$. By using Lemmas 5, 6, 7 and 8, we can know the upper bounds of the numerator in Eqn. (45) and (46) are $M(\varepsilon + e^\delta)(1 + \varepsilon)^{M-1}$ and $[M^2\varepsilon^2 + M\varepsilon + e^\delta(M^2 + M\varepsilon)](1 + \varepsilon)^{M-2}$, respectively, where $\varepsilon = e^{-K_u}$. On the other hand, by using Lemma 9, we can obtain the lower bound of the denominator as $(1 + e^\delta)[1 - e^{-c(\omega)M}](1 + \omega)^M$, where $\omega = e^{-K_l}$ and $c(\omega) = 0.5(1 - \omega)^2(1 + \omega)^{-2}$. Considering $M \gg 1$, we can make the approximation that $e^{-c(\omega)M} \approx 0$ and $(1 + e^\delta)\varepsilon/M \approx 0$. Besides, $(1 + \omega)^M \geq 1$ holds because $\omega \geq 0$. In this case, Lemma 1 can be concluded by combining the upper bound of the numerator and the lower bound of the denominator.

### B.3 H function analysis

Here, we present our analysis on the $H$ function defined in Eqn. (43). Firstly, we can have:

**Lemma 15.** $H(m, 0.5; M, t) = 2(t + 1)^{-1}$.

**Lemma 16.** $H(m, p; M, t) = H(n, \bar{p}; M, t)$.

**Lemma 17.** *As a function of $m$, $H(m, p; M, t)$ is logarithmically convex.*

*Proof.* Lemma 15 can be proved by integrating $2x^t$ on $[0, 1]$. Lemma 16 can be proved by showing that $H(n, \bar{p}; M, t)$ has the same expression as $H(m, p; M, t)$. Thus, in the following proof, we focus on Lemma 17. Fixing $p$, $M$ and $t$, we denote $\log(H)$ by $f(m)$. Then, we compute the first-order derivative as

$$H(m)f'(m) = 2^{M+1} \int_0^1 \lambda u^n (1 - u)^m x^t \mathrm{d}x \tag{47}$$

where $u = (2p - 1)x + 1 - p$ and $\lambda = \log(1 - u) - \log(u)$. Furthermore, we can solve the second-order derivative as

$$2^{-2(M+1)}H^2(m)f''(m) =$$

$$\int_0^1 g^2(x)\mathrm{d}x \int_0^1 h^2(x)\mathrm{d}x - \left( \int_0^1 g(x)h(x)\mathrm{d}x \right)^2 \tag{48}$$

where the functions $g, h : (0, 1) \to \mathbb{R}$ are defined by

$$g = \lambda\sqrt{u^n(1 - u)^m} \ , \ h = \sqrt{u^n(1 - u)^m}. \tag{49}$$

By the Cauchy-Schwarz inequality,

$$\int_0^1 g^2(x)\mathrm{d}x \int_0^1 h^2(x)\mathrm{d}x \geq \left( \int_0^1 g(x)h(x)\mathrm{d}x \right)^2 \tag{50}$$

we can know that $f''(m) \geq 0$ always holds, which concludes that $f$ is convex and $H$ is logarithmically convex. $\square$

Then, for the case that $t = 1$ and $M \gg 1$, we can further derive the following three lemmas for $H(m, p; M, 1)$:

**Lemma 18.** *The ratio between two ends satisfies*

$$\log \frac{H(0, p; M, 1)}{H(M, p; M, 1)} \approx \begin{cases} \log(M) + \epsilon(p) & p > 0.5 \\ 0 & p = 0.5 \\ -\log(M) - \epsilon(\bar{p}) & p < 0.5 \end{cases}$$

*where $\epsilon(p) = \log(2p - 1) - \log(p)$ and $\epsilon(p) = 0$ if $p = 0.5$.*

**Lemma 19.** *The lower bound can be calculated as*

$$\log H(m, p) \gtrsim \begin{cases} H(0, p) - k_l \cdot m & 2m \leq M \\ H(M, p) - k_l \cdot n & 2m > M \end{cases}$$

*where $k_l = \log\left(\max\left\{p/(\bar{p} + M^{-1}), \bar{p}/(p + M^{-1})\right\}\right)$.*

**Lemma 20.** *The upper bound can be calculated as*

$$\log H(m, p) \lesssim \begin{cases} H(0, p) - k_u \cdot m & 2m \leq M \\ H(M, p) - k_u \cdot n & 2m > M \end{cases}$$

*where $n = M - m$ and $k_u = 2\log\left(2 \cdot \max\{p, \bar{p}\}\right)$.*

*Proof.* By Lemma 15, $\log H(m, 0.5; M, 1) \equiv 0$, which proves the above three lemmas for the case that $p = 0.5$. Considering the symmetry ensured by Lemma 16, we thus focus on the case that $p > 0.5$ in the following proof and transform $H(m, p)$ into the following formulation

$$H(m, p) = \omega(p) \cdot \int_{\bar{p}}^{p} x^n (1-x)^m (x - 1 + p) \mathrm{d}x \tag{51}$$

where $\omega(p) = 2^{M+1}/(2p-1)^2$. Then, we can solve $H(0, p)$ and $H(M, p)$ as

$$\begin{aligned} H(0, p) &= \omega(p) \int_{\bar{p}}^{p} x^M (x - \bar{p}) \mathrm{d}x \\ &= \frac{(2p)^{M+1}}{(2p-1)(M+1)} - O\left(\frac{(2p)^{M+1}}{M^2}\right) \end{aligned} \tag{52}$$

$$\begin{aligned} H(M, p) &= \omega(p) \int_{\bar{p}}^{p} (1-x)^M (x - \bar{p}) \mathrm{d}x \\ &= \frac{p(2p)^{M+1}}{(2p-1)^2(M+1)(M+2)} - O\left(\frac{(2\bar{p})^{M+1}}{M+2}\right). \end{aligned} \tag{53}$$

Using the Taylor expansion of function $\log(x)$, we can calculate the ratio in Lemma 18 as

$$\log \frac{H(0, p)}{H(M, p)} = \log(M) + \log \frac{2p-1}{p} + O\left(\frac{1}{M}\right) \tag{54}$$

which concludes Lemma 18 when $M \gg 1$.

Furthermore, we can solve $H(1, p)$ as

$$\begin{aligned} H(1, p) &= \omega(p) \int_{\bar{p}}^{p} x^{M-1} (x - \bar{p}) \mathrm{d}x - H(0, p) \\ &= \frac{(2\bar{p} + M^{-1})(2p)^M}{(2p-1)(M+1)} - O\left(\frac{(2p)^{M+1}}{M^2}\right) \end{aligned} \tag{55}$$

The value ratio between $m = 0$ and $m = 1$ then satisfies

$$\log \frac{H(1, p)}{H(0, p)} = \log \frac{p}{\bar{p} + M^{-1}} + O\left(\frac{1}{M}\right). \tag{56}$$

By Rolle's theorem, there exists a $c \in [m, m+1]$ satisfying

$$\log H(1, p) - \log H(0, p) = f'(c) \tag{57}$$

where $f(m) = \log H(m, p)$. Meanwhile, Lemma 17 ensures that $f''(m) \geq 0$ always holds. Thus, we can have

$$\log H(m+1, p) - \log H(m, p) \geq \log \frac{H(1, 0)}{H(0, p)} \tag{58}$$

which concludes the first case of Lemma 19. Similarly, we compute the ratio between $m = M - 1$ and $M$ as

$$\log \frac{H(M, p)}{H(M-1, p)} = \log \frac{p}{\bar{p} + M^{-1}} + O\left(\frac{1}{M}\right). \tag{59}$$

Meanwhile, Rolle's theorem and Lemma 17 ensure that

$$\log H(m, p) - \log H(m-1, p) \leq \log \frac{H(M, 0)}{H(M-1, p)} \tag{60}$$

which concludes the second case of Lemma 19.

Lastly, we focus on the upper bound described by Lemma 20. According to the inequality of arithmetic and geometric means, $x(1-x) \leq 2^{-2}$ holds for any $x \in [0, 1]$. Thus, when $2m \leq M$ (i.e. $n \geq m$), we can have

$$H(m, p) \leq 2^{-2m} \omega(p) \cdot \int_{\bar{p}}^{p} x^{n-m} (x - 1 + p) \mathrm{d}x \tag{61}$$

where the equality only holds when $m = 0$.

$$\int_{\bar{p}}^{p} x^{n-m}(x-1+p)\mathrm{d}x = \frac{(2p-1)p^{\delta}}{\delta} + \frac{\Delta}{\delta(\delta+1)} \tag{62}$$

where $\delta = n - m + 1$ and $\Delta = \bar{p}^{\delta+1} - p^{\delta+1} < 0$. Hence,

$$\log \frac{H(m,p)}{H(0,p)} \le -2m[\log(2p) - \varepsilon(m)] + O\left(\frac{1}{M}\right) \tag{63}$$

where $\varepsilon(m) = -(2m)^{-1}[\log(n-m+1) - \log(M+1)]$. Since $\log(x)$ is a concave function, we can know that

$$\varepsilon(m) \le (M)^{-1}\log(M+1) = O\left(M^{-1}\right) \tag{64}$$

which concludes the first case in Lemma 20. Similarly, for $2m > M$ (i.e. $n < m$), we can have

$$\log \frac{H(m,p)}{H(M,p)} \le -2n[\log(2p) - \hat{\varepsilon}(n)] + O\left(\frac{1}{M}\right) \tag{65}$$

where $\hat{\varepsilon}(n) \le O(M^{-1})$. Thereby, we can conclude the second case of Lemma 20. Note that the case where $p < 0.5$ can be derived by using Lemma 16. $\square$

For the case that $t = 0$ and $M \gg 1$, using the same method as the above proof, we can derive the same lower and upper bounds as Lemmas 20 and 19. On the other hand, for $t = 0$, Lemma 18 does not hold and we can have

**Lemma 21.** $H(m,p;M,0) = H(n,p;M,0)$
*Proof.* When $t = 0$,

$$H(m,p) = 2^{M+1}(2p-1)^{-1}\int_{\bar{p}}^{p} x^n(1-x)^m \mathrm{d}x. \tag{66}$$

Then, substituting $x$ as $1 - v$ concludes Lemma 21. $\square$

## C  Proof for Theorem 1

Following the notations in Section C, when $M \gg 1$, in Eqn. (**??**), we have $\tilde{\mathbb{P}}_i = \mathbb{E}_{\mathcal{L}}(x_i + z_i)/M + O(1/M)$, where $\mathbb{E}_{\mathcal{L}}$ denotes the expectation of $\mathbb{P}(\mathcal{L}|\boldsymbol{L})$. Meanwhile, according to Chebyshev's inequality, $\mathbb{P}_i = (x_i + w_i)/M + \epsilon$, where $|\epsilon| \le_{1-\delta} O(1/\sqrt{\delta M})$ and $\delta$ is any given number in $(0, 1)$. Here, we use $a \le_{1-\delta} b$ to denote that $a$ is smaller or equal than $b$ with probability $1 - \delta$. Thus, we can calculate the upper bound of $|\tilde{\mathbb{P}}_i - \mathbb{P}_i|$ as

$$|\tilde{\mathbb{P}}_i - \mathbb{P}_i| \le_{1-\delta} \mathbb{E}_{\mathcal{L}}|w_i - z_i|/M + O(1/\sqrt{M}) \le \mathbb{E}_{\mathcal{L}}[m/M] + O(1/\sqrt{M}). \tag{67}$$

Recalling Lemma 1, we know that when $M \gg 1$,

$$\mathbb{E}[m/M] \lesssim (1+e^{\delta})^{-1}(\varepsilon + e^{\delta})(1+\varepsilon)^{M-1} \ , \ \mathbb{E}[m/M]^2 \lesssim (1+e^{\delta})^{-1}(\varepsilon^2 + e^{\delta})(1+\varepsilon)^{M-2}. \tag{68}$$

where $\varepsilon^{-1} = \prod_{i=0}^{N}(2\hat{\mathbb{P}}_i)^2$, $\delta = O[\Delta \cdot \log(M)]$ and $\Delta = \sum_{i=1}^{N}[1(\mathbb{P}_i < 0.5) - 1(\mathbb{P}_i > 0.5)]$. If $\Delta < 0$, from the definition of $\Delta$, we can know that $\Delta \le 1$. Thus, $e^{\delta} \le O(1/M)$. Furthermore, when $\prod_{i=0}^{N}(2\hat{\mathbb{P}}_i)^2 \ge M$, $\varepsilon \le M^{-1}$. Thereby,

$$\mathbb{E}\left[\frac{m}{M}\right] \lesssim \frac{C_1}{M \cdot C_2} \ , \ \mathbb{E}\left[\frac{m}{M}\right]^2 \lesssim \frac{C_1}{M^2 \cdot C_2^2} \tag{69}$$

where $C_1 = (1 + M^{-1})^M \approx e$ and $C_2 = 1 + M^{-1} \approx 1$. Based on Eqn. (69), we can know $\mathbb{E}[m/M] \lesssim O(1/M)$ and $\mathrm{Var}[m/M] \lesssim O(1/M^2)$. Again, according to Chebyshev's inequality, we can have $\mathbb{E}_{\mathcal{L}}[m/M] \le_{1-\delta} O(1/\sqrt{\delta M})$, and we can conclude Theorem 1 by taking the upper bound of $\mathbb{E}_{\mathcal{L}}[m/M]$ into Eqn. (67).

## D  Background for Reinforcement Learning

In this section, we introduce some important concepts about reinforcement learning (RL). In an RL problem, an agent interacts with an unknown environment and attempts to maximize its cumulative collected reward [9, 10]. The environment is commonly formalized as a Markov Decision Process (MDP) defined as $\mathcal{M} = \langle \mathcal{S}, \mathcal{A}, \mathcal{R}, \mathcal{P}, \gamma \rangle$. At time $t$ the agent is in state $s_t \in \mathcal{S}$ where it takes an action $a_t \in \mathcal{A}$ leading to the next state $s_{t+1} \in \mathcal{S}$ according to the transition probability kernel $\mathcal{P}$, which encodes $\mathbb{P}(s_{t+1} \mid s_t, a_t)$. In most RL problems, $\mathcal{P}$ is unknown to the agent. The agent's goal is to learn the optimal policy, a conditional distribution $\pi(a \mid s)$ that maximizes the sate's value function. The value function calculates the cumulative reward the agent is expected to receive given it would follow the current policy $\pi$ after observing the current state $s_t$

$$V^\pi(s) = \mathbb{E}_\pi \left[ \sum_{k=1}^{\infty} \gamma^k r_{t+k} \mid s_t = s \right].$$

Intuitively, it measures how preferable each state is given the current policy.

As a critical step towards improving a given policy, it is a standard practice for RL algorithms to learn a state-action value function (i.e. Q-function). Q-function calculates the expected cumulative reward if agent choose $a$ in the current state and follows $\pi$ thereafter

$$Q^\pi(s, a) = \mathbb{E}_\pi \left[ \mathcal{R}(s_t, a_t, s_{t+1}) + \gamma V^\pi(s_{t+1}) \mid s_t = s, a_t = a \right].$$

In real-world problems, in order to achieve better generalization, instead of learning a value for each state-action pair, it is more common to learn an approximate value function: $Q^\pi(s, a; \theta) \approx Q^\pi(s, a)$. A standard approach is to learn a feature-based state representation $\phi(s)$ instead of using the raw state $s$ [3]. Due to the popularity of Deep Reinforcement learning, it has been a trend to deploy neural networks to automatically extract high-level features [8, 6]. However, running most deep RL models are very computationally heavy. On contrast, static feature representations are usually light-weight and simple to deploy. Several studies also reveal that a carefully designed static feature representation can achieve performance as good as the most sophisticated deep RL models, even in the most challenging domains [4].

## E  Utility-Maximizing Strategy for Workers

**Lemma 22.** *For worker $i$, when $M \gg 1$ and $a_t > \frac{c_{i,H}}{\mathbb{P}_{i,H} - 0.5}$, if $\tilde{\mathbb{P}}_i^t \approx \mathbb{P}_i^t$, reporting truthfully ($\mathsf{rpt}_i^t = 1$) and exerting high efforts ($\mathsf{eft}_i^t = 1$) is the utility-maximizing strategy.*

*Proof.* When $M \gg 1$, we can have $\sum_j \mathsf{sc}_i(j) \approx M \cdot \tilde{\mathbb{P}}_i$. Thus, the utility of worker $i$ can be computed as

$$u_i^t \approx M \cdot a_t \cdot (\tilde{\mathbb{P}}_i - 0.5) + M \cdot b - M \cdot c_{i,H} \cdot \mathsf{eft}_i^t. \tag{70}$$

Further considering Eqn. (1) and $\mathbb{P}_L = 0.5$, if $\tilde{\mathbb{P}}_i^t \approx \mathbb{P}_i^t$, we can compute worker $i$'s utility as

$$u_i^t \approx M \cdot [a_t(2 \cdot \mathsf{rpt}_i^t - 1)(\mathbb{P}_{i,H} - 0.5) - c_{i,H}] \cdot \mathsf{eft}_i^t + M \cdot b. \tag{71}$$

Thereby, if $a_t > \frac{c_{i,H}}{\mathbb{P}_{i,H} - 0.5}$, $\mathsf{rpt}_i^t = 1$ and $\mathsf{eft}_i^t = 1$ maximize $u_i^t$, which concludes Lemma 22.  □

## F  Uninformative Equilibrium

The uninformative equilibrium denotes the case where all workers collude by always reports the same answer to all tasks. For traditional peer prediction mechanisms, under this equilibrium, all the workers still can get high payments because these mechanisms determines the payment by comparing the reports of two workers. However, the data requester only can get uninformative labels, and thus this equilibrium is undesired.

In our mechanism, when workers always reports the same answer, for example 1, our Bayesian inference will wrongly regard the collected labels as high-quality ones and calculate the estimates as

$$\tilde{\mathbb{P}}_i = \frac{M+2}{M+3} \,, \; \tilde{\tau}_{-1} = \frac{M+1}{M+2}. \tag{72}$$

If the answer is 2, our estimates are

$$\tilde{\mathbb{P}}_i = \frac{M+2}{M+3} \ , \ \tilde{\tau}_{+1} = \frac{M+1}{M+2}. \tag{73}$$

In this case, we can build a warning signal for the uninformative equilibrium as

$$\mathsf{Sig}_u = \frac{1}{N}\sum_{i=1}^{N}\log(\tilde{\mathbb{P}}_i) + \log(\max\{\tilde{\tau}_1, \tilde{\tau}_2\}) \tag{74}$$

If

$$\mathsf{Sig}_u \geq \log\frac{M+1}{M+3} \tag{75}$$

workers are identified to be under the uniformative equilibrium, and we will directly set the score in our payment rule as $0$. By doing so, we can create a huge loss for workers and push them to leave this uninformative equilibrium.

# G   Proof for Theorem 3

In our proof, if we omit the superscript $t$ in an equation, we mean that this equation holds for all time steps. Due to the one step IC, we know that, to get higher long term payments, worker $i$ must mislead our RIL algorithm into at least increasing the scaling factor from $a$ to any $a' > a$ at a certain state $\hat{s}$. Actually, our RIL algorithm will only increase the scaling factor when the state-action value function satisfies $Q^\pi(\hat{s}, a) \leq Q^\pi(\hat{s}, a')$. Eqn. (??) tells us that the reward function consists of the utility obtained from the collected labels ($F(\tilde{A}^t)$) and the utility lost in the payment ($\eta\sum_{i=1}^{N}P_i^t$). Once we increase the scaling factor, we at least need to increase the payments for the other $N-1$ workers by $M\sum_{x\neq i}\mathbb{P}_{x,H}\cdot G_{\mathcal{A}}$, corresponding to the left-hand side of the first equation in Eqn. (??).

On the other hand, for the obtained utility from the collected labels, we have

**Lemma 23.** *At any time step t, if all workers except worker i report truthfully and exert high efforts, we have $F(\tilde{A}^t) \leq F(1)$ and $F(\tilde{A}^t) \geq F(1-\psi)$, where $\psi$ is defined in Eqn. (??).*

*Proof.* In our Bayesian inference algorithm, when $M \gg 1$, the estimated accuracy $\tilde{A}$ satisfies

$$\tilde{A} \approx 1 - \mathbb{E}g(\tilde{\sigma}_j) \ , \ g(\tilde{\sigma}_j) = 1/(1+e^{|\tilde{\sigma}_j|}). \tag{76}$$

From the proof of Theorem 2, we can know that $\tilde{\mathbb{P}}_i^t \approx \mathbb{P}_i^t$. In this case, according to Eqn. (??), we can have

$$\tilde{\sigma}_j(\mathbb{P}_i) \approx \log\left(\frac{\tau_{-1}}{\tau_{+1}}\lambda_i^{\delta_{ij1}-\delta_{ij2}}\prod_{k\neq i}\lambda_H^{\delta_{kj1}-\delta_{kj2}}\right). \tag{77}$$

where $\lambda_i = \mathbb{P}_i/(1-\mathbb{P}_i)$ and $\lambda_H = \mathbb{P}_H/(1-\mathbb{P}_H)$.

We know that $\tilde{A} \leq 1.0$ holds no matter what strategy worker $i$ takes. To prove Lemma 2, we still need to know the lower bound of $\tilde{A}$. Thus, we consider two extreme cases where worker $i$ intentionally provides low-quality labels:

**Case 1**: If $\mathbb{P}_i = 0.5$, we can eliminate $\lambda_i$ from Eqn.77 because $\lambda_i = 1$. Furthermore, according to Lemma 12, we can know that $g(\tilde{\sigma}_j) < e^{\tilde{\sigma}_j}$ and $g(\tilde{\sigma}_j) < e^{-\tilde{\sigma}_j}$ both hold. Thus, we build a tighter upper bound of $g(\tilde{\sigma}_j)$ by dividing all the combinations of $\delta_{kj1}$ and $\delta_{kj2}$ in Eqn.77 into two sets and using the smaller one of $e^{\tilde{\sigma}_j}$ and $e^{-\tilde{\sigma}_j}$ in each set. By using this method, if the true label is $-1$, we can have $\mathbb{E}_{[L(j)=-1]}g(\tilde{\sigma}_j) < q_1 + q_2$, where

$$q_1 = \frac{\tau_{+1}}{\tau_{-1}}\sum_{n=K+1}^{N-1}C_{N-1}^n(\frac{1}{\lambda_H})^{n-m}\mathbb{P}_H^n(1-\mathbb{P}_H)^m$$

$$q_2 = \frac{\tau_{-1}}{\tau_{+1}}\sum_{n=0}^{K}C_{N-1}^n\lambda_H^{n-m}\mathbb{P}_H^n(1-\mathbb{P}_H)^m$$

$$n = \sum_{k\neq i}\delta_{kj(-1)} \ , \ m = \sum_{k\neq i}\delta_{kj(+1)}$$

and $K = \lfloor(N-1)/2\rfloor$. By using Lemma 13, we can thus get

$$\mathbb{E}_{[L(j)=-1]}g(\tilde{\sigma}_j) < c_\tau[4\mathbb{P}_H(1-\mathbb{P}_H)]^{\frac{N-1}{2}}.$$

where $c_\tau = \tau_{-1}\tau_{+1}^{-1} + \tau_{-1}^{-1}\tau_{+1}$. Similarly,

$$\mathbb{E}_{[L(j)=+1]}g(\tilde{\sigma}_j) < c_\tau[4\mathbb{P}_H(1-\mathbb{P}_H)]^{\frac{N-1}{2}}.$$

Thereby, $\tilde{A} > 1 - c_\tau[4\mathbb{P}_H(1-\mathbb{P}_H)]^{\frac{N-1}{2}} = 1 - \psi$.

**Case 2**: If $\mathbb{P}_i = 1 - \mathbb{P}_H$, we can rewrite Eqn.77 as

$$\tilde{\sigma}_j(1-\mathbb{P}_H) \approx \log\left(\frac{\tau_{-1}}{\tau_{+1}}\lambda_H^{x-y}\prod_{k\neq i}\lambda_H^{\delta_{kj(-1)}-\delta_{kj(+1)}}\right)$$

where $x = \delta_{ij(+1)}$ and $y = \delta_{ij(-1)}$. Since $\mathbb{P}_i = 1 - \mathbb{P}_H$, $x$ and $y$ actually has the same distribution as $\delta_{kj(-1)}$ and $\delta_{kj(+1)}$. Thus, the distribution of $\tilde{\sigma}_j(1-\mathbb{P}_H)$ is actually the same as $\tilde{\sigma}_j(\mathbb{P}_H)$. In other words, since Theorem 2 ensures $\tilde{\mathbb{P}}_i \approx \mathbb{P}_i$, our Bayesian inference algorithm uses the information provided by worker $i$ via flipping the label when $\mathbb{P}_i < 0.5$.

Thus, even if worker $i$ intentionally lowers the label quality, $\tilde{A} \geq 1 - \psi$ still holds. Considering $F(\cdot)$ is a non-decreasing monotonic function, we conclude Lemma 2. $\square$

Thereby, if Eqn. (13) is satisfied, worker $i$ will not be able to cover $Q$ value loss in the payments, and our RL algorithm will reject the hypothesis to increase the scaling factor. In this case, the only utility-maximizing strategy for worker $i$ is to report truthfully and exert high efforts.

## H  Worker Models

To demonstrate the general applicability of our mechanism, we test it under three different worker models in Section 5.2, with each capturing a different way to decide the labeling strategy. The formal description of the three models is as follows:

- **Rational** workers alway act to maximize their own utilities. Since our incentive mechanism theoretically ensures that exerting high effort is the utility-maximizing strategy for all workers (proved in Section 4), it is safe to assume workers always do so as long as the payment is high enough to cover the cost.
- **Quantal Response (QR)** workers [5] exert high efforts with the probability

$$\mathsf{eft}_i^t = \frac{\exp(\lambda \cdot u_{iH}^t)}{\exp(\lambda \cdot u_{iH}^t) + \exp(\lambda \cdot u_{iL}^t)}$$

  where $u_{iH}^t$ and $u_{iL}^t$ denote worker $i$'s expected utility after exerting high or low efforts respectively at time $t$. $\lambda$ describe workers' rationality level and we set $\lambda = 3$.
- **Multiplicative Weight Update (MWU)** workers [2] update their probabilities of exerting high efforts at every time step $t$ after receiving the payment as the following equation

$$\mathsf{eft}_i^{t+1} = \frac{\mathsf{eft}_i^t(1+\bar{u}_{\cdot H})}{\mathsf{eft}_i^t(\bar{u}_{\cdot H} - \bar{u}_{\cdot L}) + \bar{u}_{\cdot L} + 1}$$

  where $\bar{u}_{\cdot H}$ and $\bar{u}_{\cdot L}$ denote the average utilities received if exerting high efforts or low efforts at time $t$ respectively. We initialize $\mathsf{eft}_i^0$ as $0.2$ in our experiments.