[Reviews · NeurIPS 2018]

Reviewer 1



Summary: In this paper, the authors explore the problem of data collecting using crowdsourcing. In the setting of the paper, each task is a labeling task with binary labels, and workers are strategic in choosing effort levels and reporting strategies that maximize their utility. The true label for each task and workers’ parameters are all unknown to the requester. The requester’s goal is to learn how to decide the payment and how to aggregate the collected labels by learning from workers’ past answers. The authors’ proposed approach is a combination of incentive design, Bayesian inference, and reinforcement learning. At each time step, the authors use Gibbs sampling to help infer the true labels. The estimates are then fed into the reinforcement framework to decide how to choose the payments. Simulated experiments are conducted to evaluate the performance of the proposed approach. Comments: On the positive side, while the techniques used in the paper seem pretty standard individually, I like the idea of using Bayesian inference to obtain estimated ground truth for deciding how to set the payments. The overall approach also seems sound and works reasonably well in simulations. On the negative side, I have some concerns about the analysis (more comments below). Using only simulated experiments for evaluations is also not satisfactory (especially a big component of the paper is on learning worker strategies). Overall, I would still slightly lean towards accepting the paper but won’t argue for it. I have a few questions about the analysis. In Theorem 1, the authors show that the estimate of P_i (PoBC) will converge. However, the proof seems to treat P_i as a constant while it is actually a strategic choice of agent i (as described in Equation 1). This is fine if we can show that agents always exert high effort and gives truthful reports, as stated in Theorem 2. However, the proof of Theorem 2 is based on the condition that P_i already converges. This creates the chicken-and-egg problem (Theorem 1 relies on Theorem 2 to be true, and Theorem 2 relies on Theorem 1 to be true). I might have missed something. It would be helpful if the authors can clarify this. In Theorem 2 and 3, it is assumed that a_t is larger than some threshold (which is a function of workers’ cost and accuracy). However, in Algorithm 2, there is a probability of \epsilon that a_t will be randomly chosen from A. To satisfy the condition, it seems we need to make all choices in A to be larger than the threshold. Does this mean that the requester needs to know workers’ cost/accuracy in advance? Overall, the analysis on the three major components seem relatively isolated, while they influence each other. For example, when there is not enough data in the beginning, the Bayesian inference won’t be accurate, could this change workers’ strategies in the beginning and latter influence the follow-up learning (I lean towards believing this won’t happen given the simulation results, however, it is not clear from the current analysis)? In Figure 2-(a), I am surprised to see the aggregation accuracy of the proposed approach significantly outperforms variational inference even when no noise is added (variational inference has been reported to perform consistently well in the literature). If this result is consistent across many datasets, it could be worth more discussion. I think the paper would be much stronger if there are some real-world experiments, since real workers are usually not rational (or QR/MWU) and don’t just have two effort levels.

Reviewer 2



This paper considered incentive mechanism design in crowdsourcing. Compared to the existing works that focused on one-shot static solutions, this paper proposed a novel inference aided reinforcement mechanism that can learn to incentivize high quality-data sequentially without any prior assumptions. The problem itself is quite interesting and the solutions are promising. Some minor concerns are listed as follows (a) A Bayesian inference model is considered in this paper. However, Bayesian model is impractical in many applications due to memory and computation constraints in the system. Do this issues exist in this work? What is the complexity of this inference model? Also is it possible to consider a non-Bayesian model here? (b) There are some assumptions in Section 2. First, for the assumptions on the \mathbb{P}_{i, H}, \mathbb{P}_{i, L} and c_{i, H}, c_{i, L}, are the values are commonly used in the literature? Do the choices of these values have an impact on the results or not? Second, for equation (2), workers are assumed to be risk-neutral, hence, is risk- neutral an underlying assumption for all workers and data requester in this paper? (c) Dominant strategy incentive compatible (IC) and individually rational (IR) are two properties people usually have to take care of in mechanism design especially with self-interested users. From the Reinforcement Incentive Learning (RIL) algorithm, it is clear that IC has been well addressed. How the algorithm guarantee IR? Is something I missed here?

Reviewer 3



This paper builds an inference-aided reinforcement mechanism leveraging Bayesian inference for crowdsourcing, and reinforcement learning techniques to learn the optimal policy to incentivize high-quality labels. The proposed method is proved to be incentive compatible, and its benefits have been empirically shown. This paper is well written, and the technical part is sound. The idea of using reinforcement learning for crowdsourcing incentive mechanism design is interesting and novel. Different from existing solutions, it can adaptively change the payment based on the performance of workers. Moreover, it’s impressive that the proposed mechanism works well for different kind of workers, e.g. rational, QR, and MWU. I don’t have much concern about the paper’s quality, but have several questions about the details for the authors: +Line 124, you claim “the EM and VI methods tend to heavily bias toward over-estimating the 
aggregated label accuracy when the quality of labels is low. 
” Can you explain why this happened? Moreover, there exist many other label aggregation methods, such as minimax entropy estimator; will these methods bring further improvements comparing with the Bayesian estimator you used here? +You use a lot of estimated values, such as accuracy, utilities, etc., instead of the exact values when defining the MDP. Can you analyze the effect of this approximation? For example, you can show the performance under different approximating levels. +Why you choose the RTE dataset for experiments? There exist many public available benchmark datasets such as WebSearch, Age, TEMP, etc., so you at least should give some explanation about the dataset selection. + You mix data with random noise in order to simulate strategic behaviors of workers. However, the give crowdsourced labels already contain noises. So will this influence the conclusion we observed from the experiments? + For empirical analysis on RIL, the hyperparameter selection seems arbitrary. Can you give some hints on how to select these values? +It’s better to conduct some real people labeling experiments with the given mechanism, and compare the performance and efficiency with other mechanisms. After Rebuttal: Thanks for the reply, some of my concerns have been addressed. To further improve the paper, I still suggest the authors to do some real people labeling experiments.